# Genetic nurture in intergenerational transmission of substance use

Mannan Luo [1] ✉, Victória Trindade Pons [1], Nathan A. Gillespie [2,3] & Hanna M. van Loo [1]

Substance use runs in families. Beyond genetic transmission, parental genetics can indirectly influence offspring substance use through the rearing environment, known as genetic nurture. This study utilizes transmitted and non-transmitted polygenic scores to investigate genetic nurture on tobacco, alcohol, and cannabis use in up to 15,863 adults with at least one genotyped parent from Lifelines, a population-based cohort study. Genetic nurture significantly influences cigarettes per day (CPD, $\beta = 0.037$, $p_{FDR} = 0.020$) and pack-years ($\beta = 0.028$, $p_{FDR} = 0.035$), accounting for 22–26% of direct genetic transmission effects. Longitudinal analysis reveals that genetic nurture on current CPD persists across adulthood, whereas direct genetic transmission effects attenuate with age. Maternal and paternal genetic nurture are similar in magnitude. Mediation analyses indicate that genetic nurture partially operates through both parents' smoking quantity, with a stronger mediated effect through maternal smoking, particularly among daughters. These findings highlight genetic nurture as a persistent mechanism in the intergenerational transmission of smoking, operating through parental smoking behaviors.

Substance use, including tobacco, alcohol, and cannabis use, is a major public health concern that often runs in families[1,2]. Parental substance use is among the most salient familial risk factors for offspring substance use and related disorders[3–5]. To break this cycle, it is critical to understand how this risk transmits across generations. It is well established that both genetic liability and environmental influences contribute to intergenerational transmission of substance use[6–8]. However, disentangling these pathways is challenging because many putatively environmental risk factors, including parental substance use, are genetically influenced, creating confounding through gene-environment correlation[9,10].

A powerful methodological advance for disentangling intergenerational pathways is to construct polygenic scores for the alleles transmitted ($PGS_T$) and not transmitted ($PGS_{NT}$) from parents to offspring[11,12]. The influence of non-transmitted parental alleles ($PGS_{NT}$) on offspring outcomes quantifies genetic nurture, a process whereby parents' genotypes influence offspring outcomes indirectly through the

rearing environment, independent of direct genetic transmission captured by $PGS_T$[11,12]. For instance, parents with a higher genetic predisposition for substance use may use substances more frequently in the home or model such behaviors[2,13], thereby increasing the offspring's risk of substance use beyond their own inherited genetic liability.

Despite growing evidence for genetic nurture effects on substance use[14,15], key gaps remain. First, existing research has been predominantly limited to adolescence and young adulthood, often with a narrow focus on a specific substance. As such, it remains unclear whether genetic nurture effects generalize across different substances and persist into adulthood, even after offspring leave the family household and parental influences diminish[16]. Second, while it is often hypothesized that genetic nurture effects attenuate over time[15,17], no study to date has directly examined the temporal effects of genetic nurture across adulthood using repeated measures of substance use. Third, the common practice of aggregating maternal and paternal PGSs can obscure parent-of-origin effects, as both direct genetic

[1]Department of Psychiatry, University Medical Center Groningen, University of Groningen, Groningen, the Netherlands. [2]Virginia Institute for Psychiatric and Behavioral Genetics, Virginia Commonwealth University, Richmond, VA, USA. [3]Genetic Epidemiology, QIMR Berghofer, Brisbane, QLD, Australia. ✉e-mail: m.luo@umcg.nl

inheritance[18] and genetic nurture[11,19] may differ by parent. Finally, although the presence of genetic nurture implies environmental mediation, the specific mechanisms remain underexplored.

In light of these gaps, we leveraged data from Lifelines, a large population-based cohort study, to address three key questions: (i) Do genetic nurture effects manifest across different substances (smoking initiation, cigarettes per day, pack-years, daily alcohol intake, and cannabis initiation) and persist into adulthood? And if so, do these effects remain stable or change with age? (ii) Do maternal and paternal genetic nurture differ in magnitude (i.e., parent-of-origin effects)? (iii) To what extent is genetic nurture mediated by parental substance use, and does this mediation differ between mothers and fathers? By addressing these questions, we aim to elucidate the temporal dynamics and mechanisms underlying the intergenerational transmission of substance use.

## Results

### Population characteristics

We utilized data from a total of 19,233 genotyped adult offspring with at least one parent genotyped from a Dutch general population cohort study, Lifelines, comprising 15,966 parent-offspring pairs and 3267 mother-father-offspring trios. Of these, up to 15,863 participants (mean age$_{baseline}$ = 31.66 years; 61.9% female), consisting of 13,411 pairs and 2452 trios with available substance use data, completed assessments for tobacco and alcohol use at baseline (2006–2013) and cannabis use during the second wave (2014–2017). Descriptive statistics for these participants are presented in Table 1 and Supplementary Figs 1 and 2.

We examined baseline differences between offspring with complete trio data (both parents genotyped, $n = 2454$) and those with only one genotyped parent ($n = 13,417$). Offspring from trios were significantly younger, had a lower proportion of females, and reported lower lifetime cigarettes per day and pack-years, with all standardized mean differences (SMDs) ranging from 0.07 to 0.18. These modest differences underscore the importance of including both family structures in our analyses to enhance the generalizability of findings. Correlations among PGSs and offspring outcomes are provided in Supplementary Fig. 3.

### Genetic nurture effects across substance use

We utilized a validated haplotype-based approach[20] to construct PGS$_T$ and PGS$_{NT}$ for smoking initiation (SmkInit), cigarettes per day (CPD), alcoholic drinks per week (DPW), and cannabis initiation (CanInit). We combined maternal and paternal PGS$_T$ and PGS$_{NT}$ for statistical power (see Supplementary Information). Linear mixed regression models were used to examine the overall effects of PGS$_T$ and PGS$_{NT}$ (Table 2). Following Kong et al.[11], we estimated the effect of direct genetic transmission ($\beta_{DGT}$) by subtracting the effect of PGS$_{NT}$ from that of PGS$_T$ (i.e., $\beta_T - \beta_{NT}$). This approach isolates true genetic transmission by removing the influence of genetic nurture, given that PGS$_T$ may influence offspring directly through genetic inheritance and indirectly via genetic nurture.

As expected, each PGS$_T$ was significantly associated with its corresponding substance use outcomes (binary: OR = 1.27–1.85; continuous: $\beta$ = 0.124–0.202). For genetic transmission effects, $\beta_{DGT}$ was 0.17 for CPD, and 0.11 for pack-years and daily alcohol intake. For genetic nurture effects, parental PGS$_{NT\_CPD}$ showed relatively smaller but significant associations with offspring CPD ($\beta_{NT}$ = 0.037) and pack-years ($\beta_{NT}$ = 0.028) after false discovery rate (FDR) correction. These genetic nurture effects equaled approximately 22.4% and 25.9% of $\beta_{DGT}$ for CPD and pack-years, respectively. While parental PGS$_{NT\_DPW}$ was significantly associated with offspring daily alcohol intake, this effect did not survive FDR correction. No significant associations were found between parental PGS$_{NT}$ and either smoking initiation or cannabis initiation.

Because offspring from complete parent-offspring trios reported lower CPD and pack-years than those from pairs, we performed sensitivity analyses by fitting all regression models separately in each subsample. The effect estimates showed similar magnitudes with overlapping 95% confidence intervals (Supplementary Table 1), suggesting that our main findings are robust to these differences in tobacco use.

Given the substantial genetic correlations across substance use traits and disorders (SUDs)[21], we conducted cross-trait analyses using a broad PGS$_{SUD}$ to further examine the specificity of genetic nurture effects (Supplementary Table 2). Consistent with our primary analyses using trait-specific PGSs, PGS$_{T\_SUD}$ was significantly associated with all substance use outcomes (ORs = 1.16–1.34 for binary traits; $\beta$s = 0.08–0.12 for continuous traits, all $p < 0.001$), confirming broad direct genetic transmission across the substance use spectrum. In contrast, PGS$_{NT\_SUD}$ showed no significant associations with any outcome, which may reflect either trait-specific genetic nurture or reduced sensitivity for detecting genetic nurture with a broad cross-trait PGS.

**Table 1 | Descriptive characteristics of Lifelines participants, including the subsamples of adult offspring from genotyped family parent-offspring pairs and trios**

| | All | | Pairs | | Trios | | p* |
|---|---|---|---|---|---|---|---|
| | N | M ± SD or n (%) | N | M ± SD or n (%) | N | M ± SD or n (%) | |
| Age (years) at assessment | 15871 | 31.67 ± 8.61 | 13417 | 31.86 ± 8.70 | 2454 | 30.59 ± 8.02 | <0.001 |
| Sex, female | 15871 | 9827 (61.9) | 13417 | 8376 (62.4) | 2454 | 1451 (59.1) | 0.002 |
| Tobacco use | | | | | | | |
| Smoking initiation, yes | 15853 | 6560 (41.4) | 13403 | 5620 (41.9) | 2450 | 940 (38.4) | 0.001 |
| Cigarette per day | 5972 | 10.24 ± 6.00 | 5134 | 10.34 ± 6.06 | 838 | 9.59 ± 5.57 | 0.001 |
| Pack-years | 6276 | 7.03 ± 6.73 | 5382 | 7.19 ± 6.85 | 894 | 6.04 ± 5.88 | <0.001 |
| Alcohol use | | | | | | | |
| Daily alcohol intake (grams/day) | 15863 | 6.74 ± 8.46 | 13411 | 6.73 ± 8.47 | 2452 | 6.81 ± 8.41 | 0.682 |
| Cannabis use | | | | | | | |
| Cannabis initiation, yes | 9097 | 2066 (22.7) | 7608 | 1710 (22.5) | 1489 | 359 (24.1) | 0.180 |

N Sample size, M Mean, SD Standard deviation.
*Chi-square test for binary outcomes and t-test for continuous outcomes when comparing the pairs versus trios on substance use. Pack-years of smoking were calculated by multiplying the amount smoked per day (of different types of tobacco products, including cigarettes/roll-ups, cigarillos, cigars, grams of pipe tobacco) by the number of years the person has smoked. All tests were two-sided.

**Table 2 | Regression coefficients of parental transmitted, non-transmitted polygenic scores, and direct genetic effects on offspring substance use in adulthood**

| | N | $\beta_T$ (SE)/OR | PGS$_T$ 95% CI | p | p$_{FDR}$ | $\beta_{NT}$ (SE)/OR | PGS$_{NT}$ 95% CI | p | p$_{FDR}$ | DGT $\beta_T$ - $\beta_{NT}$ |
|---|---|---|---|---|---|---|---|---|---|---|
| PGS$_{SmkInit}$ | | | | | | | | | | |
| Smoking initiation | 15853 | 1.853[a] | 1.767, 1.944 | <0.001 | <0.001 | 0.989[a] | 0.950, 1.031 | 0.612 | 0.612 | – |
| PGS$_{CPD}$ | | | | | | | | | | |
| Cigarettes per day | 5972 | 0.202 (0.013) | 0.177, 0.226 | <0.001 | <0.001 | 0.037 (0.013) | 0.012, 0.062 | 0.004 | 0.020 | 0.165 |
| Pack-years | 6276 | 0.136 (0.011) | 0.114, 0.157 | <0.001 | <0.001 | 0.028 (0.011) | 0.006, 0.049 | 0.014 | 0.035 | 0.108 |
| PGS$_{DPW}$ | | | | | | | | | | |
| Daily alcohol intake | 15863 | 0.124 (0.007) | 0.110, 0.139 | <0.001 | <0.001 | 0.016 (0.008) | 0.001, 0.031 | 0.034 | 0.057 | 0.108 |
| PGS$_{CanInit}$ | | | | | | | | | | |
| Cannabis initiation | 9097 | 1.274[a] | 1.188, 1.365 | <0.001 | <0.001 | 1.033[a] | 0.967, 1.104 | 0.328 | 0.410 | – |

Mixed-effects regression models include up to 15,863 adult offspring with at least one parent genotyped and available data on substance use outcomes. Age and sex were included as covariates, and sibling effect was controlled for by including family ID as a random effect into models.
$\beta_T$ and $\beta_{NT}$ standardized coefficients of the polygenic scores calculated for the transmitted and non-transmitted alleles, respectively, when they are analyzed jointly. DGT $\beta_T$–$\beta_{NT}$ estimated effect of genetic transmission of the polygenic score. *SE* standard error, *OR* odds ratio, *95% CI* 95% confidence interval, *p* unadjusted p value, *p$_{FDR}$* p value after false discovery rate (FDR) correction for multiple comparisons. *SmkInit* smoking initiation, *CPD* cigarettes per day, *DPW* alcoholic drink per week, *CanInit* cannabis initiation.
[a]the estimated odds ratio for smoking initiation and lifetime cannabis use from logistic regression models.

**Table 3 | Longitudinal associations between transmitted and non-transmitted polygenic scores and current cigarettes per day across adulthood**

| Predictor | β (SE) | 95% CI | p |
|---|---|---|---|
| Fixed effects | | | |
| PGS$_{T\_CPD}$ | 0.120 (.012) | 0.097, 0.143 | <0.001 |
| PGS$_{NT\_CPD}$ | 0.026 (.012) | 0.003, 0.049 | 0.028 |
| Age | −0.040 (0.007) | −0.054, −0.025 | <0.001 |
| Sex (male) | 0.085 (.024) | 0.039, 0.132 | <0.001 |
| Birth year | −0.249 (0.065) | −0.377, −0.121 | <0.001 |
| Interaction terms | | | |
| PGS$_{T\_CPD}$ × Age | −0.004 (0.001) | −0.006, −0.001 | 0.002 |
| PGS$_{NT\_CPD}$ × Age | −0.0004 (0.001) | −0.003, 0.002 | 0.761 |

Results from a linear mixed-effects model examining whether the associations between polygenic scores and current cigarettes per day (CPD) change with age. The sample included repeated measures within individuals (*n* = 6885) nested within families (*n* = 5755), totaling 9725 observations across three waves. Age was centered at the baseline mean (33.0 years). The model included PGS$_{T\_CPD}$ × Age and PG$_{NT\_CPD}$ × Age interaction terms to test age-varying effects, and adjusted for sex, birth year, and measurement wave. We additionally included random intercepts and slopes for individuals to account for repeated measures, and random intercepts for families to account for sibling relatedness. No adjustments were made for multiple comparisons, as all interaction coefficients were estimated within a single model.
β standardized coefficient, *SE* standard error, *95% CI* 95% confidence interval, *p* unadjusted p value.

Overall, these findings indicate significant genetic nurture effects specifically on lifetime smoking quantity (CPD and pack-years). In contrast, tobacco or cannabis initiation and daily alcohol intake were predominantly influenced by direct genetic transmission, with minimal contributions from genetic nurture.

**Temporal effects of genetic nurture across adulthood**
To examine whether genetic nurture effects remain stable or change over time, we analyzed repeated measures of current CPD across three waves using linear mixed-effects models with PGS×age interactions (Table 3 and Supplementary Fig. 4). Results revealed distinct temporal patterns for genetic transmission versus genetic nurture. The PGS$_{T\_CPD}$ × age interaction was significantly negative ($\beta = -0.004$, $p = 0.002$), indicating that genetic transmission effects attenuated with age. In contrast, the PGS$_{NT\_CPD}$ × age interaction was non-

significant ($\beta = -0.0003$, $p = 0.79$), suggesting genetic nurture effects remain stable across adulthood. These findings indicate that while the influence of one's own genetics on smoking quantity decreases over time, the environmental influences shaped by parental genetics persist throughout adulthood.

**Parent-of-origin effects**
Given the significant genetic nurture effects observed for lifetime CPD and pack-years, we further examined parent-specific effects using separate maternal and paternal PGS$_{T\_CPD}$ and PGS$_{NT\_CPD}$ (Table 4). Structural equation modeling (SEM) was used to simultaneously estimate maternal and paternal PGS$_T$ and PGS$_{NT}$ effects, while accounting for potential genetic assortative mating for tobacco use[22].

Both maternal and paternal PGS$_{T\_CPD}$ significantly predicted offspring smoking outcomes, with Wald tests showing no significant difference in effect magnitude between parents (CPD: $\Delta\chi^2 = 3.35$, $p = 0.07$; pack-years: $\Delta\chi^2 = 2.49$, $p = .12$). Genetic nurture effects showed similar patterns for both parents across outcomes. For pack-years, effects of maternal ($\beta = 0.027$, $p = 0.047$) and paternal ($\beta = 0.034$, $p = 0.043$) PGS$_{NT\_CPD}$ were both significant and statistically equivalent ($\Delta\chi^2 = 0.01$, $p = 0.91$). For CPD, effect magnitudes were nearly identical (maternal $\beta = 0.039$; paternal $\beta = 0.036$), though only the maternal effect reached statistical significance ($p = 0.008$ vs. $p = 0.057$), likely due to reduced power in the smaller CPD sample. Correlations between maternal and paternal PGSs were non-significant (Supplementary Table 3), indicating minimal genetic assortative mating for smoking quantity. This suggests our genetic nurture estimates were unlikely to be inflated by genetic similarity between parents.

To assess power for detecting parent-of-origin effects on smoking quantity (Supplementary Table 4 and Fig. 5), we conducted Monte Carlo simulations (1000 replications) across plausible effect sizes defined by their observed estimates and 95% CIs. At the observed effect sizes, power was moderate for transmitted parent-of-origin effects (44.7% for CPD, 37.1% for pack-years). However, if the maternal-paternal difference were larger in magnitude, corresponding to the lower confidence bounds ($\Delta\beta = -0.067$ for CPD, −0.056 for pack-years), power would be high (≥95%). Power for non-transmitted effects was low to moderate (5.1 – 68.6%) across the CI range. Therefore, the absence of significant parent-of-origin effects should be interpreted with caution. Substantially larger samples would be required to

**Table 4 | Parent-of-origin effects on offspring smoking quantity: comparison of transmitted and non-transmitted polygenic scores for cigarettes per day split by paternal and maternal haplotypes**

| | Maternal PGS$_{T\_CPD}$ | | | Maternal PGS$_{NT\_CPD}$ | | | Paternal PGS$_{T\_CPD}$ | | | Paternal PGS$_{NT\_CPD}$ | | |
|---|---|---|---|---|---|---|---|---|---|---|---|---|
| | β (SE) | 95% CI | p | β (SE) | 95% CI | p | β (SE) | 95% CI | p | β (SE) | 95% CI | p |
| Cigarettes per day | 0.124 (0.012) | 0.100, 0.148 | <0.001 | 0.039 (0.015) | 0.010, 0.068 | 0.008 | 0.155 (0.013) | 0.131, 0.180 | <0.001 | 0.036 (0.019) | −0.001, 0.073 | 0.057 |
| Pack-years | 0.082 (0.011) | 0.060, 0.103 | <0.001 | 0.027 (0.013) | 0.000, 0.052 | 0.047 | 0.107 (0.011) | 0.085, 0.128 | <0.001 | 0.034 (0.017) | 0.001, 0.067 | 0.043 |

Structural equation modeling was used to assess the effects of maternal and paternal transmitted (PGS$_T$) and non-transmitted (PGS$_{NT}$) polygenic scores on offspring smoking outcomes. All models adjust for offspring sex and age, and account for sibling relatedness using clustered robust standard errors. As the analyses used a full information maximum likelihood (FIML) approach to handle missing data, retaining all available data from the full genotyped family sample (N = 19,233), there was no listwise N for the sample. Standardized coefficients (β), cluster-robust standard errors (SE), 95% confidence intervals (CIs) and unadjusted p value are reported. No adjustments were made for multiple comparisons given the high correlation between outcome measures (cigarettes per day and pack-years), which would make such correction overly conservative for non-independent tests.

adequately test for parent-of-origin differences, particularly for non-transmitted effects.

## Mediation by parental tobacco use

To explore the extent to which parental tobacco use mediated genetic nurture effects identified above, and whether mediation differed between parents, we fitted a joint SEM that simultaneously estimated maternal and paternal mediation pathways while accounting for genetic and phenotypic assortative mating.

As shown in Fig. 1A, maternal CPD significantly mediated effects of both PGS$_T$ ($\beta_{mediation}$ = 0.026) and PGS$_{NT}$ ($\beta_{mediation}$ = 0.029) on offspring CPD, with paternal mediation effects being weaker but also significant (transmitted $\beta_{mediation}$ = 0.008; non-transmitted $\beta_{mediation}$ = 0.009). Similar patterns emerged for pack-years (Fig. 1B), with maternal pathways showing significant mediation for both transmitted and non-transmitted effects, while paternal pathways did not reach significance.

Analysis of parent-of-origin effects in mediation (Supplementary Table 5) revealed that maternal smoking mediated genetic nurture effects more strongly than paternal smoking across both outcomes (transmitted: $\Delta\beta_{mediation\_CPD}$ = 0.018, $p$ = 0.003; $\Delta\beta_{mediation\_pack-years}$ = 0.012, $p$ = 0.004; nontransmitted: $\Delta\beta_{mediation\_CPD}$ = 0.022, $p$ < 0.001; $\Delta\beta_{mediation\_pack-years}$ = 0.016, $p$ < 0.001). These findings suggest that genetic nurture operates through parent-specific mediation pathways for smoking quantity, with maternal pathways playing a more predominant role than paternal smoking.

We further explored whether parental mediation effects differed between daughters and sons using multi-group SEM. Sex-stratified analyses revealed that maternal mediation was significantly stronger than paternal mediation in daughters (transmitted: $\Delta\beta_{mediation}$ = 0.019; non-transmitted: $\Delta\beta_{mediation}$ = 0.023), but not in sons (transmitted: $\Delta\beta_{mediation}$ = 0.013; non-transmitted: $\Delta\beta_{mediation}$ = 0.017). However, when comparing maternal and paternal pathways separately across sex groups, neither maternal pathways nor paternal pathways differed significantly between daughters and sons (Supplementary Table 6).

## Discussion

Leveraging a large, population-based cohort with genotyped parent-offspring trios and pairs, this study provides the first comprehensive investigation of genetic nurture effects across multiple substances and across adulthood, revealing important insights into the persistence and mechanisms underlying intergenerational transmission of substance use. We highlight four key findings. First, genetic nurture significantly influenced smoking quantity (cigarettes per day and pack-years), but not smoking initiation, cannabis initiation, or daily alcohol intake, demonstrating substance specificity. In contrast, direct genetic transmission played a substantial role across all measured outcomes. Second, while direct genetic effects on smoking quantity diminished over time, genetic nurture effects remained consistent throughout

adulthood. Third, parent-of-origin analyses revealed that the magnitudes of genetic nurture effects on smoking quantity were nearly identical for mothers and fathers. Finally, parental smoking quantity mediated genetic nurture effects, with maternal smoking having a stronger impact on offspring smoking than paternal smoking, particularly among daughters.

Our results demonstrate the specificity of genetic nurture in substance use: significant effects were observed for smoking quantity (cigarettes per day, pack-years) but not for initiation of smoking or cannabis, and daily alcohol intake. This pattern aligns with evidence that genetic and environmental contributions vary across different dimensions of substance use[6,7,23–25], such as initiation, quantity, and dependence. The observed genetic nurture for smoking quantity reflects passive gene-environment correlation (rGE)[26], whereby parental genotypes shape the rearing environment in which smoking is frequently modeled. Lifetime CPD and pack-years capture sustained smoking patterns involving long-term regulation and reinforcement[27,28]. These cumulative behaviors are more likely influenced by the enduring family environment shaped by parental genetics.

In contrast, the absence of genetic nurture for tobacco or cannabis initiation, and daily alcohol intake may reflect the environmental factors operating independently of parental genetics, the transient nature of these behaviors, and measurement characteristics. Tobacco and cannabis initiation represent discrete behavioral events occurring within peer networks and immediate social contexts[29–31], while daily alcohol consumption fluctuates with social situations and life events (e.g., being a college student, celebrations or holidays)[32,33]. Rather than passive rGE (i.e., genetic nurture), these behaviors may be driven by other types of rGE, such as active rGE, whereby individuals select environments (e.g., peer groups, social settings) that align with their own genetic predispositions, facilitating initial substance use[34,35]. Methodologically, dichotomous measures of initiation may lack sensitivity to detect subtle genetic nurture effects, despite large sample sizes (N = 15,853 for smoking initiation). Similarly, our short-term assessment of alcohol use (past 30 days) may not adequately capture the stable, long-term drinking patterns[36] that are more likely to reflect early family influences. This is consistent with evidence that genetic nurture is more salient for problematic alcohol use[14] than for normative consumption[15], highlighting its primary role in sustained, heavy or problematic use.

Another novel contribution of our study is the longitudinal analysis demonstrating that genetic nurture effects on smoking quantity persist across adulthood, while the effects of direct genetic transmission attenuate with age. The persistence of genetic nurture across the adult lifespan aligns with developmental models[37] emphasizing long-term parental influences on offspring outcomes. The age-related attenuation of direct genetic effects might reflect the growing influence of non-familial environments (e.g., reduced smoking due to

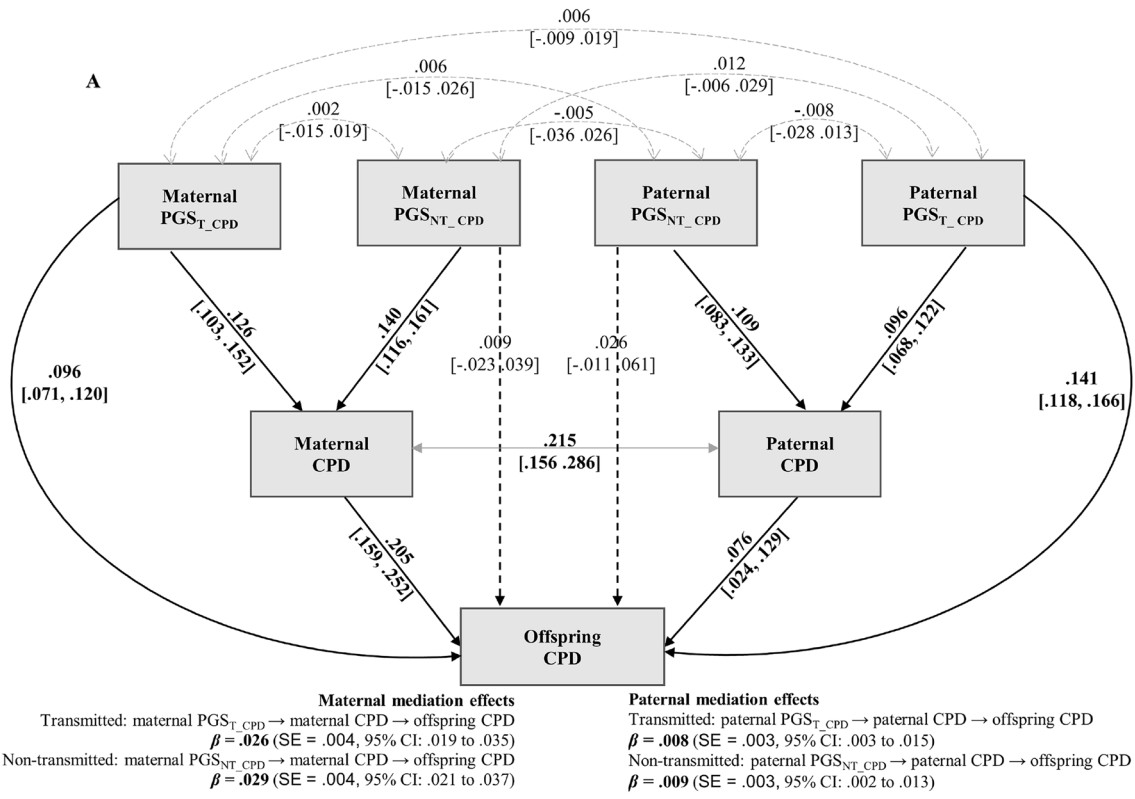

Maternal mediation effects
Transmitted: maternal PGS$_{T\_CPD}$ → maternal CPD → offspring CPD
**β = .026** (SE = .004, 95% CI: .019 to .035)
Non-transmitted: maternal PGS$_{NT\_CPD}$ → maternal CPD → offspring CPD
**β = .029** (SE = .004, 95% CI: .021 to .037)

Paternal mediation effects
Transmitted: paternal PGS$_{T\_CPD}$ → paternal CPD → offspring CPD
**β = .008** (SE = .003, 95% CI: .003 to .015)
Non-transmitted: paternal PGS$_{NT\_CPD}$ → paternal CPD → offspring CPD
**β = .009** (SE = .003, 95% CI: .002 to .013)

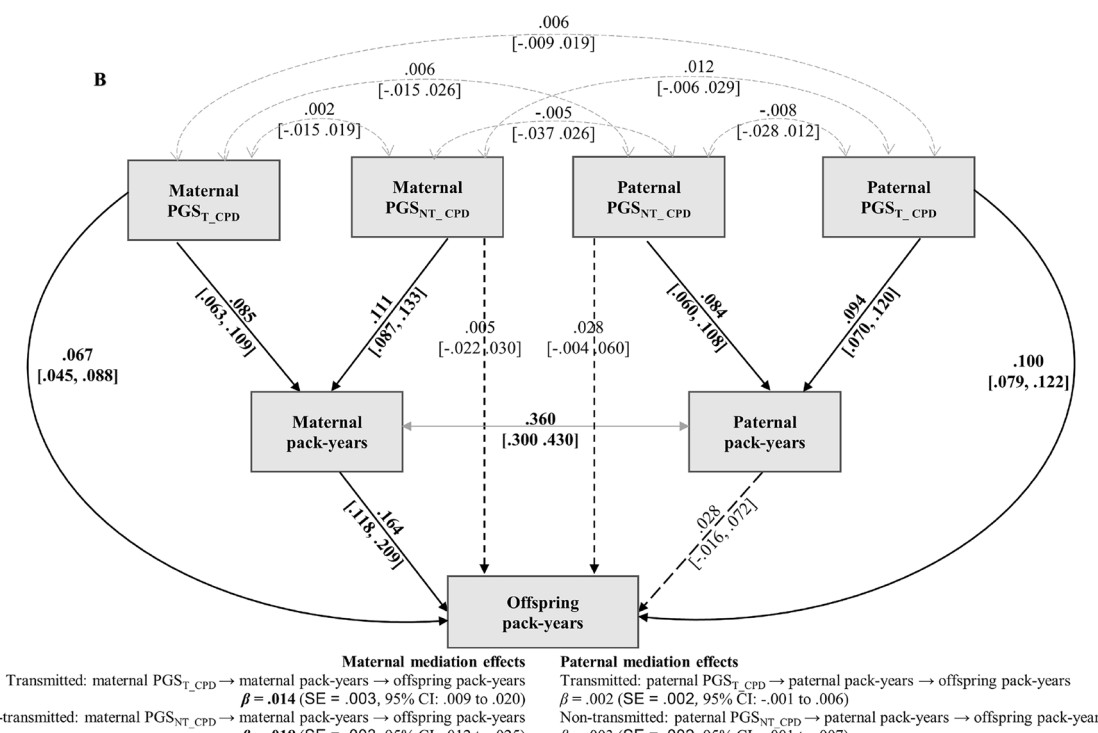

Maternal mediation effects
Transmitted: maternal PGS$_{T\_CPD}$ → maternal pack-years → offspring pack-years
**β = .014** (SE = .003, 95% CI: .009 to .020)
Non-transmitted: maternal PGS$_{NT\_CPD}$ → maternal pack-years → offspring pack-years
**β = .018** (SE = .003, 95% CI: .012 to .025)

Paternal mediation effects
Transmitted: paternal PGS$_{T\_CPD}$ → paternal pack-years → offspring pack-years
**β = .002** (SE = .002, 95% CI: -.001 to .006)
Non-transmitted: paternal PGS$_{NT\_CPD}$ → paternal pack-years → offspring pack-years
**β = .003** (SE = .002, 95% CI: -.001 to .007)

health concerns or lifestyle changes) that suppress genetic predispositions as individuals age. These divergent temporal patterns are consistent with previous research suggesting that genetic and environmental effects on substance use are dynamic and may shift over time[38,39].

Our findings advance understanding of how genetic nurture operates across the lifespan. A prior study found genetic nurture effects on smoking at age 24 but not at 29 using separate cross-sectional analyses[15], whereas our within-individual repeated measures design provides greater power to detect subtle, persistent effects that might be obscured in single-time-point comparisons[40]. This underscores the critical importance of longitudinal designs with repeated measures to elucidate how genetic transmission and genetic nurture unfold over time. Future studies

**Fig. 1 | Mediation analysis using structural equation modeling: maternal and paternal smoking quantity as mediators of the associations between transmitted (PGS_T) and non-transmitted (PGS_NT) polygenic scores and offspring smoking outcomes. All models adjust for offspring sex and age.** Solid lines indicate significant pathways, and dashed lines represent non-significant pathways. Double-headed gray arrows indicate covariances between parental polygenic scores and phenotypes, reflecting assortative mating. Reported effects are standardized estimates with bootstrapped 95% CI. Bold indicates a significant mediation effect. **A** Cigarettes per day. Total effect of maternal transmitted: $\beta_{total} = 0.122$, SE = 0.012, 95% CI [0.098, 0.145] and non-transmitted: $\beta_{total} = 0.038$, SE = 0.015, 95% CI [0.008, 0.067]. Total effect of paternal transmitted: $\beta_{total} = 0.150$, SE = 0.012, 95% CI [0.127, 0.172] and non-transmitted: $\beta_{total} = 0.033$, SE = 0.018, 95% CI [−0.003, 0.068]. **B** Pack-years. Total effect of maternal transmitted: $\beta_{total} = 0.081$, SE = 0.011, 95% CI [0.060, 0.102] and non-transmitted: $\beta_{total} = 0.023$, SE = 0.013, 95% CI [−0.003, 0.048]; Total effect of paternal transmitted: $\beta_{total} = 0.102$, SE = 0.011, 95% CI [0.082, 0.124] and non-transmitted: $\beta_{total} = 0.031$, SE = 0.017, 95% CI [−0.002, 0.063]. No adjustments were made for multiple comparisons, as all mediation parameters were estimated with bootstrapping.

replicating these temporal dynamics in independent samples are warranted.

Although maternal and paternal genetic nurture effects were equal in magnitude overall, our mediation analysis revealed that the specific pathway through observable smoking behavior was stronger for mothers. This stronger maternal mediation likely reflects mothers' primary role in shaping the day-to-day home environment and child-rearing practices during critical developmental periods[41]. Mothers typically spend more time with children than fathers[42,43], which may amplify the impact of maternal smoking through increased opportunities for behavioral modeling[44,45] consistent with social learning theory[46]. Additionally, mothers may exert unique influences through prenatal smoking exposure[47], which has been linked to neurobiological changes in offspring, including alterations in early brain development[48] and epigenetic patterns[49], thereby increasing the risk of later substance use[50,51]. Fathers' genetic nurture effects were also partly mediated by their smoking quantity, though these effects were weaker than maternal pathways. This suggests that paternal genetic nurture likely operates through additional pathways not included in our model, such as psychiatric disorders[52] and father-child relationship[53]. Future research should explore alternative mediation pathways to gain a more comprehensive understanding of the environmental mechanisms underlying genetic nurture.

Additional mediation analyses by offspring sex revealed that maternal mediation significantly exceeded paternal mediation among daughters but not among sons. This finding may reflect same-sex parental modeling processes[54], whereby daughters more strongly identify with and emulate maternal behaviors compared to paternal behaviors. However, the non-significant result in sons should be interpreted cautiously, as it likely reflects reduced statistical power in the smaller male sample rather than a true absence of effect, given the overlapping confidence intervals and comparable point estimates between sexes. Therefore, these exploratory findings warrant replication in larger samples.

The strengths of our study include the large population-based cohort, a novel approach integrating genotypic data from parent-offspring trios and pairs, and the use of diverse substance use measures. Moreover, we employed longitudinal modeling with repeated measures at three time points and structural equation modeling to investigate parent-of-origin effects in genetic nurture and mediation pathways. However, several limitations remain. First, our sample included only Dutch participants of European descent, potentially limiting generalizability. Second, PGSs explain only a small proportion of genetic liability to substance use, and the modest genetic nurture effect sizes observed reflect this limitation. Third, as with most observational studies, statistical power to detect small effects in interaction (e.g., $PGS_{NT} \times Age$) and subgroup analyses (e.g., sex differences in mediation) was limited. Additionally, fewer fathers than mothers had both genotypic and phenotypic data available, which may have further constrained power for paternal analyses. Finally, retrospective self-reports on substance use, such as pack-years, may be subject to recall bias or underreporting[55]. However, measures like cigarettes per day and pack-years have demonstrated strong reliability and validity in assessing lifetime smoking exposure[56].

In conclusion, this study demonstrates that genetic nurture effects on substance use persist across adulthood and operate through parental smoking quantity, with maternal smoking exerting a stronger influence. These findings highlight the enduring role of family environments shaped by parental genetics in determining offspring smoking outcomes across the lifespan. The persistence of these effects underscores the potential for family-based interventions, particularly those targeting maternal smoking reduction, to yield lasting benefits for preventing smoking among offspring into adulthood.

## Methods
### Ethics
The Lifelines cohort study is conducted according to the principles of the Declaration of Helsinki and in accordance with the research code of the University Medical Center Groningen (UMCG). The Lifelines protocol has been approved by the UMCG Medical Ethics Committee under number 2007/152. Written informed consent was obtained from all individuals included in the study. Participants received no financial compensation.

### Participants
Lifelines is a multi-disciplinary prospective population-based cohort study examining, in a unique three-generation design, the health and health-related behaviors of 167,729 persons living in the North of the Netherlands. It employs a broad range of investigative procedures in assessing the biomedical, socio-demographic, behavioral, physical and psychological factors which contribute to the health and disease of the general population, with a special focus on multi-morbidity and complex genetics[57,58]. Data collection was accomplished at three general assessments, along with additional assessments. The baseline assessment took place between 2006 and 2013, followed by a second wave between 2014 and 2017, and a third wave between 2019 and 2023. The design and sample characteristics of Lifelines have been described in detail elsewhere[57,58]. This study included 15,863 adult offspring (61.9% female; mean age = 31.67 years, SD = 8.61) with at least one genotyped parent. Sex was determined by self-report and was included as a covariate in all analyses.

### Measurements
**Substance use.** Substance use was measured by self-report questionnaires at baseline for adult offspring and their parents, except for cannabis use, which was measured at the second wave[59]. Smoking initiation was defined as having smoked for one year or longer, with the question "Have you ever smoked for a full year in your lifetime?" Participants who smoked for less than a year were not considered smokers. For participants who met the smoking initiation criterion, lifetime smoking behavior was assessed using two complementary measures. Cigarettes per day (CPD) represented the lifetime average number of cigarettes smoked daily, assessed for both current and former smokers. Pack-years were calculated by multiplying the average amount smoked per day (including cigarettes, cigarillos, cigars, and pipes) by the number of years the person smoked in their lifetime (1 pack-year equals 20 cigarettes per day for one year). Pack-years provide a cumulative indicator of lifetime smoking dose across multiple tobacco products. These lifetime measures maximize phenotypic

variance and statistical power by incorporating complete smoking histories of both current and former smokers[60]. However, they cannot assess whether genetic nurture effects change over time. To examine potential temporal patterns directly, we analyzed current CPD ("How many cigarettes do you currently smoke per day?"), with former smokers coded as zero. Current CPD was repeatedly assessed at three waves (Wave 1: mean age = 33.0 years, SD = 8.47, range = 18−67; Wave 2: mean age = 38.9 years, SD = 9.28, range = 19−72; Wave 3: mean age = 44.8 years, SD = 8.99, range = 20−70), enabling longitudinal analysis of the stability or change in genetic nurture effects throughout adulthood.

Alcohol use was assessed with a food frequency questionnaire developed by Wageningen University[61]. Two questions referred to the frequency and quantity of alcohol consumed in the past month: "How often did you drink alcoholic drinks in the past month?" (ranging from "not this month" to "6–7 days per week"), and "On days that you drank alcohol, how many glasses did you drink on average?" (from "1" to "12 or more"). These questions were split up for different alcoholic groups (beer, alcohol-free beer, red wine/rose, white wine, sherry, distilled wine, other alcoholic beverages). Based on these questions, an average daily alcohol consumption in grams per day was calculated[62]. This composite index of daily alcohol intake provides a more comprehensive measure of overall alcohol use than a single measure of frequency (e.g., number of drinking days per month) or quantity (e.g., glasses per day).

Cannabis initiation was defined using two questions: (i) "Have you ever used drugs?", and if yes, (ii) "Have you ever used cannabis, such as weed, marijuana, hashish?". The answer categories were recoded to ever (1) versus never (0) used cannabis.

**Genotyping and imputation.** A total of 79,988 participants were genotyped across three batches in Lifelines. Quality control (QC) of markers and samples was performed separately per batch. Detailed pre-imputation QC criteria are described in Supplementary Information.

In brief, markers that were duplicated and monomorphic, markers with a low call rate or low minor allele frequency, and markers that deviated significantly from the Hardy−Weinberg equilibrium were removed. Post QC data from each array was imputed through the Sanger Imputation Service with the Haplotype Reference Consortium v1 panel. We selected overlapping imputed markers with quality scores ≥0.8 across arrays to create a common set of markers for all genotyped parents and offspring in any arrays. Samples with a low call rate, heterozygosity outliers or mix-ups on sex and familial relationship were filtered out. Samples were restricted to participants of European ancestry, determined through principal component analysis with the 1000 Genomes reference, to control for population stratification.

**Non-transmitted alleles inference.** We applied our newly developed haplotype-based approach to differentiate transmitted and non-transmitted alleles in genotyped parent-offspring pairs and trios[20]. By including parent-offspring pairs, rather than restricting the analysis to trios, this approach mitigates potential selection bias and improves statistical power. The development and validation of this method have been described in detail elsewhere[20]. Briefly, we used SHAPEIT5 to estimate haplotypes including pedigree information[63]. Offspring haplotypes were then compared to parental haplotypes using tiles of 150 adjacent markers on each chromosome. The best match between the parent and offspring tiles, taking recombination spots into account, was used to determine which parental tiles were transmitted to the offspring. The remaining non-transmitted alleles were recorded in a separate dataset, and for parent-offspring pairs, the non-transmitted alleles of the parent who was not genotyped were set as missing. This method was validated by comparison with standard software in parent-offspring trios and found a concordance rate for the non-

transmitted alleles of 99.8%. Furthermore, the identification of non-transmitted alleles was confirmed to be unaffected by missing parental data through simulations of pairs from trios.

**Polygenic scores.** We calculated $PGS_T$ and $PGS_{NT}$ based on summary statistics from previous genome-wide association studies (GWAS) for smoking initiation[64], cigarettes per day[64], drink per week (alcohol consumption)[64], and cannabis initiation[65] (Supplementary Table 7). These GWAS were chosen because they were based on the largest sample sizes currently available for each corresponding phenotype in Lifelines. We additionally constructed a cross-trait PGS for substance use disorders ($PGS_{SUD}$) derived from the multivariate GWAS of Hatoum et al.[66], which captures shared liability across problematic alcohol use, problematic tobacco use, cannabis use disorder, and opioid use disorder. This score has been shown to predict a broad range of substance use and externalizing outcomes[66,67], supporting its relevance for investigating cross-trait genetic effects.

To increase the variance explained by each PGS, SNP effects were re-weighted using the auto setting from LDpred2 (bigsnpr, version 1.12.21)[68], a Bayesian method that adjusts the effect estimates from GWAS summary statistics by incorporating trait-specific genetic architecture (e.g., SNP-based heritability and polygenicity measured as the fraction of causal variants) and linkage disequilibrium (LD) data from UK Biobank reference panel for European ancestry[69]. For each offspring, PGSs were created based on transmitted and non-transmitted datasets. To estimate overall genetic nurture and genetic transmission effects, parental $PGS_T$ and $PGS_{NT}$ were defined as the sum of the PGS based on paternal and maternal transmitted and non-transmitted haplotypes, respectively. The value of the missing $PGS_{NT}$ in parent-offspring pairs was imputed with the average $PGS_{NT}$ of the observed parents (Supplementary Information). To estimate parent-of-origin effects, we separated four maternal and paternal $PGS_T$ and $PGS_{NT}$, respectively. To control for population structure and batch effects across arrays, we standardized PGS residuals within each array after regressing out the first ten genetic principal components.

### Statistical analysis

All analyses were conducted in R (version 4.2.1)[70]. All statistical tests were two-sided unless otherwise specified. We applied a stepwise approach in which genetic nurture effects had to be statistically significant to continue to the next analysis.

### Genetic nurture and genetic transmission of substance use

First, we used mixed-effects regression models to examine associations between parental $PGS_T$ and $PGS_{NT}$ with offspring substance use outcomes, including smoking initiation, smoking quantity (cigarettes per day, pack-years), daily alcohol intake, and cannabis initiation. Continuous outcomes were analyzed using mixed-effects linear regression with the *lmerTest* (version 3.1-3)[71], while dichotomous outcomes were analyzed using mixed-effects logistic regression with the *GLMMadaptive* (version 0.8-5)[72]. Each model was specified as follows: $Y_i = \text{Intercept}_Y + \beta PGS_T + \beta PGS_{NT} + \text{sex} + \text{age} + 1|\text{Family}ID + e_i$. Family ID was included as a random effect (intercept) to account for the relatedness among siblings, along with sex and age as covariates. False discovery rate (FDR) corrections[73] (5 tests, α = 0.05) were applied to control for multiple testing across the two mixed-effects logistic regression models (smoking initiation, cannabis initiation) and three mixed-effects linear regression models (cigarettes per day, pack-years, and daily alcohol intake).

To investigate longitudinal associations between PGSs and repeated measures of current smoking quantity (CPD), we applied linear mixed-effects models with *lmerTest*[71], including $PGS_T \times$ age and $PGS_{NT} \times$ age interaction terms to assess whether direct genetic transmission and genetic nurture effects change over time. The model was adjusted for the following covariates: offspring sex, age at

measurement, birth year, and data collection wave. Additionally, we included a random intercept for families to account for sibling relatedness, and a random intercept on the subject level to account for repeated measures of the outcome (current CPD). We further added a random slope for age to account for individual differences in the rate of change over time beyond what was captured by the fixed effects, allowing the association between time and the outcome to vary randomly across individuals. Random intercepts and slopes were specified as orthogonal (uncorrelated) to aid in model identification. For comparison, we also fit a model without interaction terms to estimate the average effect of $PGS_T$ and $PGS_{NT}$ across all three waves, assuming timing-invariant genetic effects (Supplementary Information and Supplementary Table 8).

### Parent-of-origin effects

If genetic nurture effects remained significant after FDR, we examined parent-of-origin effects on offspring's substance use outcomes using structural equation modeling (SEM) in Lavaan (version 0.6–12)[74]. SEM enabled us to handle missing data with Full Information Maximum Likelihood (FIML)[75], which utilizes all available data to estimate parameters and standard errors without imputing missing values. For each SEM model, $Y_i = \text{Intercept}_Y + \beta PGS_{T\_mother} + \beta PGS_{NT\_mother} + \beta PGS_{T\_father} + \beta PGS_{NT\_father} + \text{sex} + \text{age} + e_i$, all paths and covariances will be freely estimated. Family ID was included as a clustering variable to adjust for sibling-relatedness. To assess whether the maternal and paternal effects differed significantly, we compared the equality of standardized regression coefficients using a, via *lavTestWald* function in Lavaan.

### Mediation pathways via parental substance use

We conducted mediation analysis using SEM in Lavaan to assess the extent to which parental substance use mediates the associations of $PGS_T$ and $PGS_{NT}$ with offspring substance use outcomes. Both maternal and paternal mediation analyses examined two pathways: (i) transmitted mediation, where parental $PGS_T$ influences parental smoking, which then affects offspring outcomes, and (ii) nontransmitted mediation, where parental $PGS_{NT}$ influences parental smoking and subsequently offspring outcomes. All models controlled for offspring sex and age. To evaluate parent-of-origin effects in mediation, we compared the mediated effect magnitudes to assess whether the absolute size of the mediated pathway via parental substance use differs between mothers and fathers.

To test for offspring sex differences in mediation pathways, we fitted a multi-group mediation SEM with offspring sex as the grouping variable[74], modeling maternal and paternal mediation concurrently for daughters and sons. This approach allows formal testing of: (i) within-sex contrasts: whether mediation via maternal substance use is significantly larger than via paternal substance use within daughters and within sons, and (ii) between-sex contrasts: whether the strength of the mediation pathway from the same parent (e.g., maternal genetic nurture) differs significantly between daughters and sons.

FIML was used to handle missing data and reduce the likelihood of biased parameter estimates. Standardized mediation effects were estimated via bootstrapping with 1000 replications, with statistical significance determined by 95% bootstrap CIs excluding zero.

### Reporting summary

Further information on research design is available in the Nature Portfolio Reporting Summary linked to this article.

## Data availability

Individual-level data from the Lifelines Cohort are available under restricted access due to ethical requirements and privacy regulations protecting participant confidentiality. Researchers can apply to use the Lifelines data through the online application system. More information about how to request Lifelines data and the conditions of use can be found on their website (https://www.lifelines.nl/researcher/how-to-apply). GWAS summary statistics used to construct polygenic scores were obtained from the following publicly available sources: smoking initiation, cigarettes per day and drinks per week from the GWAS & Sequencing Consortium of Alcohol and Nicotine use (GSCAN; https://doi.org/10.13020/przg-dp88); substance use disorder from the Psychiatric Genomics Consortium (https://www.med.unc.edu/pgc/).

## Code availability

Statistical analyses were performed in R (version 4.2.1) using LDpred2(bigsnpr, version 1.12.21), lmerTest (version 3.1-3), GLMMadaptive (version 0.8-5), and lavaan (version 0.6-12). Analysis code is available on GitHub: https://github.com/mannanluo/GeneticNurture_SU.

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

## Acknowledgements

This work was supported by grants from the United States National Institutes of Health, National Institute on Drug Abuse (R01DA052453 to N.A.G. and H.V.L.). The work of H.V.L. was supported by a Veni grant from the Talent Program of the Netherlands Organization of Scientific Research (NWO-ZonMW 09150161810021 to H.V.L.). The Lifelines initiative has been made possible by subsidy from the Dutch Ministry of Health, Welfare and Sport, the Dutch Ministry of Economic Affairs, the University Medical Center Groningen (UMCG), Groningen University and the Provinces in the North of the Netherlands (Drenthe, Friesland, Groningen). We acknowledge the services of the Lifelines Cohort Study, the contributing research centers delivering data to Lifelines, and all the study participants. We also thank Prof. Jean-Baptiste Pingault for his helpful comments.

## Author contributions

M.L., N.A.G., and H.V.L. contributed to the conceptualization and designed the study. V.T.P. contributed to transmitted and non-transmitted alleles data preparation. M.L. contributed to the preparation of phenotypic and genetic data and the statistical analysis. M.L. wrote the manuscript with input from V.T.P., N.A.G., and H.V.L. N.A.G. and H.V.L. acquired funding. All authors critically interpreted the results, reviewed the manuscript, and approved the final version.

## Competing interests

The authors declare no competing interests.
