## [Transparent Peer Review File · Nature Communications]

Genetic Nurture in Intergenerational Transmission of Substance Use

Corresponding Author: Dr Mannan Luo

Version 0:

Reviewer comments:

Reviewer #1

(Remarks to the Author)

The authors have addressed all the issues I raised in the previous revision, and I believe the manuscript has improved substantially. In my opinion, this manuscript is a true masterclass in genetics. The field of genetic correlations and genetic nurture has not been studied as thoroughly as it deserves, and the authors have done an excellent job addressing these questions.

I have only one minor comment. The authors report that the $PGST_CPD \times age$ interaction is significantly negative, suggesting that genetic transmission effects attenuate with age, whereas the $PGSNT_CPD \times age$ interaction is non-significant, indicating that genetic nurture effects remain stable across adulthood. I believe that a figure illustrating these interactions would greatly improve clarity and interpretability. In particular, without a graphical representation, it is difficult to fully assess the nature and extent of the reported attenuation (for example, attenuation to which level or across which age range). It is possible that the transmitted genetic effect converges toward the stable effect observed for genetic nurture, but this cannot be easily evaluated from the table alone. A visualization could help readers better understand the developmental dynamics underlying these findings.

(Remarks on code availability)

Reviewer #2

(Remarks to the Author)

The authors have comprehensively and thoughtfully addressed my concerns. Recommend accept.

(Remarks on code availability)
